# Analysis of University Students' Mental Health from the Perspective of Occupational Harmony

Yijun Liu[1]*, Ruiqi She[2], Jin Xing[3]

1 Department of Rehabilitation Medicine, Peking University First Hospital, Beijing, China, 2 Institute for Health Systems Engineering, College of Engineering, Peking University, Beijing, China, 3 Department of Rehabilitation Medicine, Beijing Hospital, National Center of Gerontology; Institute of Geriatric Medicine, Chinese Academy of Medical Sciences, Beijing, China,

* yijunliu@pku.edu.cn

## Abstract

### Background

Mental health problems are common among university students. Based on the Model of Occupational Harmony, we investigated the relationship between occupational engagement and mental health among Chinese university students.

### Methods

A mixed methods approach was adopted. A total of 734 Chinese university students responded to online questionnaires: a socio-demographic questionnaire, the Depression Anxiety Stress Scale, the WHOQOL-BREF, and a self-designed Occupational Harmony Questionnaire. Individual interviews with 11 university students and a focus group of four students provided qualitative data.

### Results

Anxiety, depression, and stress were reported by 11.9%, 9.4%, and 2.9% of the participants, respectively. The level of occupational harmony was negatively correlated with depression, anxiety, and stress ($p < .001$). The DASS-21 score was significantly different among students who spent different amounts of time on the following activities: exercise, sleep, rest, reflective and consulting activities, study, work, and leisure and entertainment ($P < .05$). In the context of the COVID-19 pandemic, the key occupational factors contributing to university students' mental health problems included excessive mental engagement and insufficient physical engagement, excessive active engagement and insufficient quiet engagement, insufficient social engagement, excessive or insufficient *Gong* (productive) engagement, and insufficient *De* (virtuous) engagement, *Zhi* (rational) engagement, and *Kang* (health maintenance) engagement, lack of coherence within the family, school, social systems, as well as inadequate schedule planning and time management.

**Data availability statement:** All relevant data are within the paper and its Supporting Information files.

**Funding:** National High Level Hospital Clinical Research Funding (Youth Clinical Research Project of Peking University First Hospital) 2024YC10

**Competing interests:** NO authors have competing interests

## Conclusions

Occupational disharmony is related to university students' mental health problems. Occupational therapy can promote university students' mental well-being by enhancing occupational harmony.

## Introduction

University students are a key population segment for determining the economic growth and development of a society. However, they are also at high risk for distress and mental health problems [1]. Globally, the high prevalence of mental disorders among university students has become a significant and growing public health issue [2]. In a survey conducted by the World Health Organization (WHO), 31.4% (95% *CI* 30.2–32.6%) of college students screened positive for at least one common mental disorder [3].

Among the research on university students' mental health, most studies adopted a biomedical perspective using psychiatric symptomatology to investigate the "mental health" status [3–5]. Measurement tools commonly used include the General Anxiety Disorder-7 (GAD-7), the Patient Health Questionnaire (PHQ-9), and the Symptom Checklist-90 (SCL-90), etc., which can identify people who manifest mental illness symptoms. However, it should be noted that mental health is not simply an absence of mental illness. According to the definition by the WHO, mental health refers to "a state of well-being, in which an individual realizes his or her own abilities, can cope with the normal stresses of life, can work productively and is able to make a contribution to his or her community" [6, p. 12].

Two implicit problems exist with using psychiatric symptomatology to study university students' mental health. First, the multifaceted meaning of "mental health" has not been fully explored as it was only used as a euphemism for "mental illness" [7]. Second, as mental health can be seen as a spectrum [8], students with mild distress but were absent from mental illness symptoms may not be identified. Therefore, it is imperative to investigate mental health from a more positive, functional, and holistic perspective.

The profession of occupational therapy (OT) possesses knowledge and expertise in how people's participation in everyday life shapes their health and well-being [9]. As a central concept in OT, the term "occupational engagement" can offer a functional and performance-based lens to explore mental health in the context of everyday living. Engagement in occupation refers to the "performance of occupations as the result of choice, motivation, and meaning within a supportive context" [10, p. 5], including both objective and subjective aspects of a person's experience. Previous studies provided evidence that a balance of occupational engagement could be beneficial for university students' mental well-being [11, 12]. In contrast, disruptions to occupational engagement and balance were associated with poor mental health [13].

As cultural values and traditions shape how occupational engagement [14] and mental health (6) are conceptualized across contexts, a Chinese occupational therapy model, the Model of Occupational Harmony (MOHar) [15, 16], was adopted to analyze the mental health of Chinese university students from an occupational lens in this study. The MOHar was built upon traditional Chinese culture and Chinese scholars' Human Complex Systems Theory [17], which provided a conceptual framework for understanding how the orchestration of everyday occupations relates to health and well-being. Notably, the MOHar addressed that occupational engagement was driven by human consciousness [15] and thus assumed an inherent relationship between occupational engagement and mental health.

 

The term "occupational harmony" indicated harmonious human-environment transactions through occupational engagement. It was attained when a person achieved their ultimate goal of a satisfactory life [15]. The model proposed that occupational harmony can be characterized as complex equilibria among three pairs of two-sided occupational characteristics (i.e., physical and mental engagement, quiet and active engagement, individual and social engagement), harmony among five-dimensional occupational engagement (i.e., *De* [virtuous] engagement, *Zhi* [rational] engagement, *Gong* [productive] engagement, *Ai* [emotional] engagement, *Kang* [health maintenance] engagement) and coherence across multiple levels (i.e., ontosystem, small-system, medium-system, large-system, chronosystem) of human-environment transactions, which were essential to health and wellbeing.

This study used the MOHar as a theoretical base and had three aims. First, to examine university students' mental health from the perspective of occupational harmony. It was hypothesized that a higher level of occupational harmony would be negatively associated with depression, anxiety, and stress symptoms. Second, to analyze the key occupational factors causing university students' mental distress. It was hypothesized that being over-, under-, or even un-engaged by occupations with a particular characteristic can lead to occupational disharmony and mental health problems. Third, as the MOHar was a theoretical model recently developed, we aimed to generate empirical evidence for the MOHar and explore its fit in analyzing occupational engagement and mental health among university students. We hypothesized that occupational harmony is related to a higher level of mental well-being and quality of life. In contrast, occupational disharmony would result in mental health problems.

## Methods

### Design

A mixed method approach [18] was chosen to answer the research aims. The MOHar was used both deductively and inductively in collecting and analyzing data. In the quantitative phase, a cross-sectional online survey was conducted to identify the level of occupational harmony and its relationship to mental health and quality of life. In the qualitative phase, individual interviews and a focus group were performed to explore how harmonious occupational engagement affects mental health from the life experience of participants.

### Participants

Participants were recruited virtually through the WeChat social media platform. Inclusion criteria included undergraduate or graduate students enrolled in the university at the time of the study who were able to communicate in Mandarin. Students in a state of leave of absence or suspension were excluded. In the quantitative phase, convenience sampling and snowball sampling were used with 734 participants included. In the qualitative phase, participants were selected from the same sample using purposeful sampling and convenience sampling. Eleven students participated in individual interviews and four students participated in a focus group discussion. Ethical approval was obtained from the ethics committee of Peking University First Hospital (Approval number: 2020Study104) and all participants agreed to the written informed consent prior to participation. Fig 1 shows the flow chart of study population selection.

### Data collection and procedures

All data collection occurred from November 22, 2020 to December 30, 2020. Quantitative data was collected via a self-report survey through the Baishuyun online data collection platform. A total of 792 survey responses were received. Duplicate responses from the same IP address,

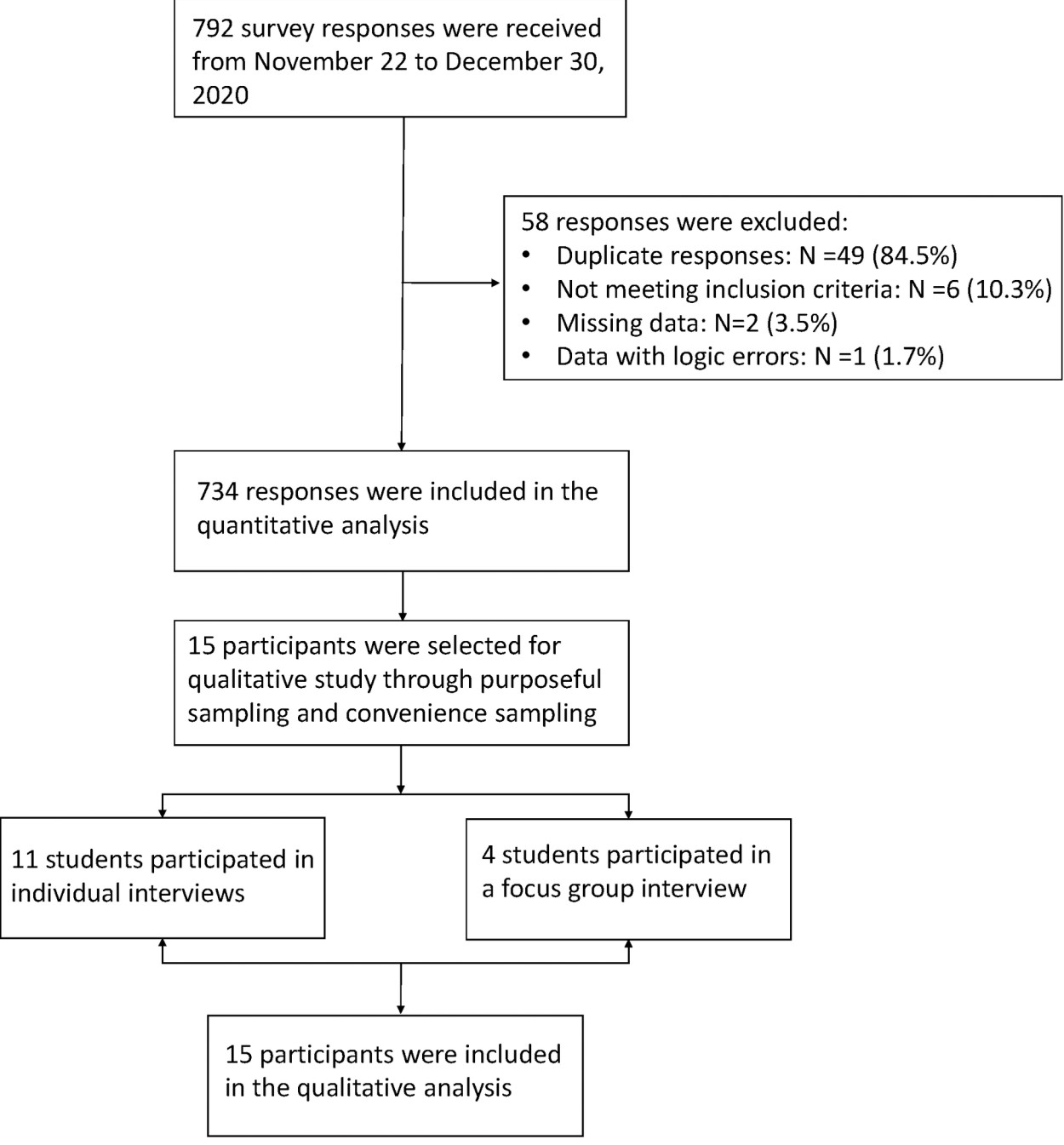

**Fig 1. Flow chart of study population selection.**

respondents who did not meet the inclusion or exclusion criteria, and answers with obvious logic errors were eliminated to ensure the authenticity and reliability of the data. Qualitative data were collected from semi-structured interviews and a focus group discussion conducted virtually via the Tencent video conference. Both visual and audio recordings were obtained. Each individual interview was 40-100 minutes long and the focus group lasted 86 minutes. After the interview, the recordings were transcribed into text data using the Sogo transcription service. Field notes and reflective journals were completed by two researchers (YL and RS).

## Instruments and measures

A structured survey containing informed consent, a socio-demographic questionnaire, the Depression Anxiety Stress Scale (DASS-21), the World Health Organization Quality of Life-BREF (WHOQOL-BREF), and a self-designed Occupational Harmony Questionnaire (OHQ), was used to collect quantitative data.

**Socio-demographic questionnaire.** The socio-demographic questionnaire sought the participants' background information such as gender, age, ethnicity, type of degree, major, monthly family income, and previous mental health issues.

**The depression anxiety stress scale (DASS-21).** The Chinese version of DASS-21 [19] was used to measure symptoms of mental illness. This instrument consists of 21 items including three subscales: 7 items each for depression, anxiety, and stress with a four-point Likert scale ranging from 0 ("never") to 3 ("always") [20]. The total sub-scores range from 0 to 42 and are categorized into normal, mild, moderate, severe, and extremely severe. In this study, Cronbach's alpha for the depression, anxiety, and stress subscales were 0.89, 0.80, and 0.84, respectively, and the overall DASS-21 scale was found to have good reliability (Cronbach's alpha = 0.94).

**The world health organization quality of life-BREF (WHOQOL-BREF).** The Chinese version of WHOQOL-BREF [21] was used to assess the quality of life (QoL). This scale contains 26 items assessing QoL in four domains: physical health, psychological health, social relations, and the environment with a five-point Likert scale ranging from 1 to 5. Transformed scores for each domain were from 0 to 100. In this study, the WHOQOL-BREF scale demonstrated good reliability (Cronbach's alpha = 0.91).

**The occupational harmony questionnaire (OHQ).** The OHQ was designed by the research team based on the MOHar to assess the objective and subjective aspects of occupational engagement, respectively. The questionnaire underwent three rounds of revision, including feedback from 10 university students, two occupational therapists, and five university teachers, cognitive interviewing [22] with two university students, and a pilot study conducted with 36 participants.

The objective subscale consists of 17 items to collect the behavior patterns of occupations across the five dimensions. An example of a question is "What is the average time you spend studying every day (including attending classes, conducting research, and internships)?" Participants responded by selecting a time category they fitted into. The subjective subscale consists of 25 items, assessing self-perceived occupational harmony levels. Participants were asked to respond on a five-point Likert scale (1 = completely inconsistent, 5 = completely consistent) to statements like "Generally speaking, I am very satisfied with my current life." Scores were summed up and converted to a transformed score ranging from 1 to 100. A higher score indicated a higher perceived level of occupational harmony. In this study, the OHQ subjective subscale was found to have good reliability (Cronbach's alpha = 0.91).

## Data analysis

**Quantitative analysis.** Of the 792 survey responses, 734 participants were included in the quantitative analysis. The Statistical Package for the Social Sciences (SPSS) (Version 26.0) was used for data analysis. The normality of the data was analyzed using the Shapiro-Wilk test. The demographic data were analyzed using descriptive statistics. Criterion validity of the OHQ subjective subscale was evaluated by the correlation between the level of occupational harmony and quality of life, using the Pearson correlation coefficient. The correlation between the level of occupational harmony and mental health was analyzed using the Spearman correlation coefficient. Comparisons of mental health status among students with different

occupational engagement profiles were analyzed using the Kruskal-Wallis *H* test. The Bonferroni method was used for post-hoc analysis. Significance was evaluated at levels of .05.

**Qualitative analysis.** A combination of deductive and inductive approaches was used in analyzing qualitative data. First, three main themes (i.e., two-sided occupational characteristics, five-dimensional occupational engagement, and multiple levels of human-environment transactions) and subthemes were developed deductively from the MOHar (See the S1 Table for details). Then, meaningful quotes from the data were assigned to the predetermined themes. New codes describing mental health status (e.g., anxious, stressed, depressive) also emerged from the data. Based on the above analysis, a profile of each participant's occupational engagement and mental health status was developed and the relationship between occupational engagement and mental health was further analyzed using the MOHar. Next, similarities and differences in participants' occupational engagement and mental health status were identified by comparing all profiles, generating common occupational factors contributing to mental health problems. All analyses were conducted by two researchers (YL and RS) with multiple discussions. The quotes were translated into English using forward translation.

## Results

### Characteristics of participants

Among the 734 participants, 518 (70.6%) were females and 576 (78.4%) were aged 19-24 years. Fifteen of them participated in the interviews and a focus group discussion. Table 1 and Table 2 show participant characteristics in the quantitative and qualitative phase of this study, respectively. Detailed profiles of 15 participants in the qualitative study were included in the S2 Table.

### Quantitative results

**Occupational engagement status.** In the OHQ objective subscale, we investigated university students' engagement in 10 occupations across five dimensions. Fig 2 shows the percentage of participants' average time-use in five-dimensional occupational engagement. The median score of the OHQ subjective subscale was 72.8 (IQR 64.0-79.2).

**Occupational harmony and quality of life.** The level of occupational harmony demonstrated significant positive correlations with the four domains of QoL: physical health (r = 0.618, *P* < .001), psychological health (r = 0.668, *P* < .001), social relations (r = 0.512, *P* < .001), and environment (r = 0.503, *P* < .001), showing occupational harmony is positively related to the quality of life and the OHQ subjective scale has a good criterion validity.

**Occupational harmony and mental health.** Among the 734 participants, 116 (15.8%) had depressive, anxiety, or stress symptoms. The prevalence rates were 9.4%, 11.8%, and 2.8%, respectively. The level of occupational harmony showed significant negative correlations with depression (r = -0.628, *P* < .001), anxiety (r = -0.500, *P* < .001), and stress (r = -0.553, *P* < .001). Table 3 shows the relationships between OHQ subjective subscale scores and DASS-21 scores.

**Occupational factors associated with mental health problems.** Participants who studied less than 2 hours a day, slept less than 6 hours a day, rested less than 40 minutes per week, or had unhealthy eating patterns, demonstrated higher scores for depression, anxiety, and stress than the other groups. Participants who studied or worked more than 10 hours a day had a higher score for anxiety. Participants who did not participate in any reflective and consulting activities had a higher score for depression. Participants who exercised over 150 minutes per week had a lower score for depression. Students spending time on leisure and entertainment for less than 1 hour had lower scores for depression, anxiety, and stress than the other groups.

**Table 1. Participant characteristics in the quantitative study (N = 734).**

| Characteristics | No. (%) |
|---|---|
| Gender | |
| Male | 216 (29.4) |
| Female | 518 (70.6) |
| Age (years) | |
| ≤18 | 58 (7.9) |
| 19-21 | 371 (50.5) |
| 22-24 | 205 (27.9) |
| 25-27 | 61 (8.3) |
| ≥28 | 39 (5.3) |
| Ethnicity | |
| Han | 689 (93.9) |
| Minorities | 45 (6.1) |
| Types of degree | |
| Undergraduate | 493 (67.2) |
| Graduate | 241 (32.8) |
| Major | |
| Medical | 223 (30.4) |
| Non-medical | 511 (69.6) |
| Location of university | |
| Mainland China | 705 (96.0) |
| Out of China | 29 (4.0) |
| Monthly family income (Yuan) | |
| <1000 | 48 (6.5) |
| 1001-3000 | 231 (31.5) |
| 3001-5000 | 172 (23.4) |
| 5001-7000 | 115 (15.7) |
| 7001-10000 | 79 (10.8) |
| ≥10001 | 89 (12.1) |
| Prior mental health issues | |
| Yes | 29 (4.0) |
| No | 705 (96.0) |

Table 4 shows the difference in DASS-21 scores of university students with different time-use in various occupations.

## Qualitative findings

Participants' occupational engagement and mental health status were analyzed according to the MOHar. Three main themes and their subthemes were used to guide data analysis as follows.

**Theme 1: Two-sided occupational characteristics and mental health.** The MOHar proposed that equilibria between three pairs of two-sided occupational characteristics were essential for health and well-being [15]. The state of equilibrium is determined by the individual's self-perception and favorable time allocation. In this study, we found that excessive mental and active engagement and insufficient social engagement were common among university students, which contributed to mental health problems.

**Table 2. Participant characteristics in the qualitative study (N = 15).**

| Characteristics | Median | Range |
|---|---|---|
| Age | 21 | 19-28 |
| OHQ subjective subscale score | 70.4 | 47.2-79.2 |
| **Characteristics** | **No.** | **Percentage** |
| DASS-21 score | | |
| Positive | 5 | 33.3% |
| Negative | 10 | 66.7% |
| Gender | | |
| Male | 3 | 20.0% |
| Female | 12 | 80.0% |
| Types of degree | | |
| Undergraduate | 10 | 66.7% |
| Graduate | 5 | 33.3% |
| Prior mental health issues | | |
| Yes | 2 | 13.3% |
| No | 13 | 86.7% |

OHQ: The Occupational Harmony Questionnaire, DASS-21: The Depression Anxiety Stress Scale.

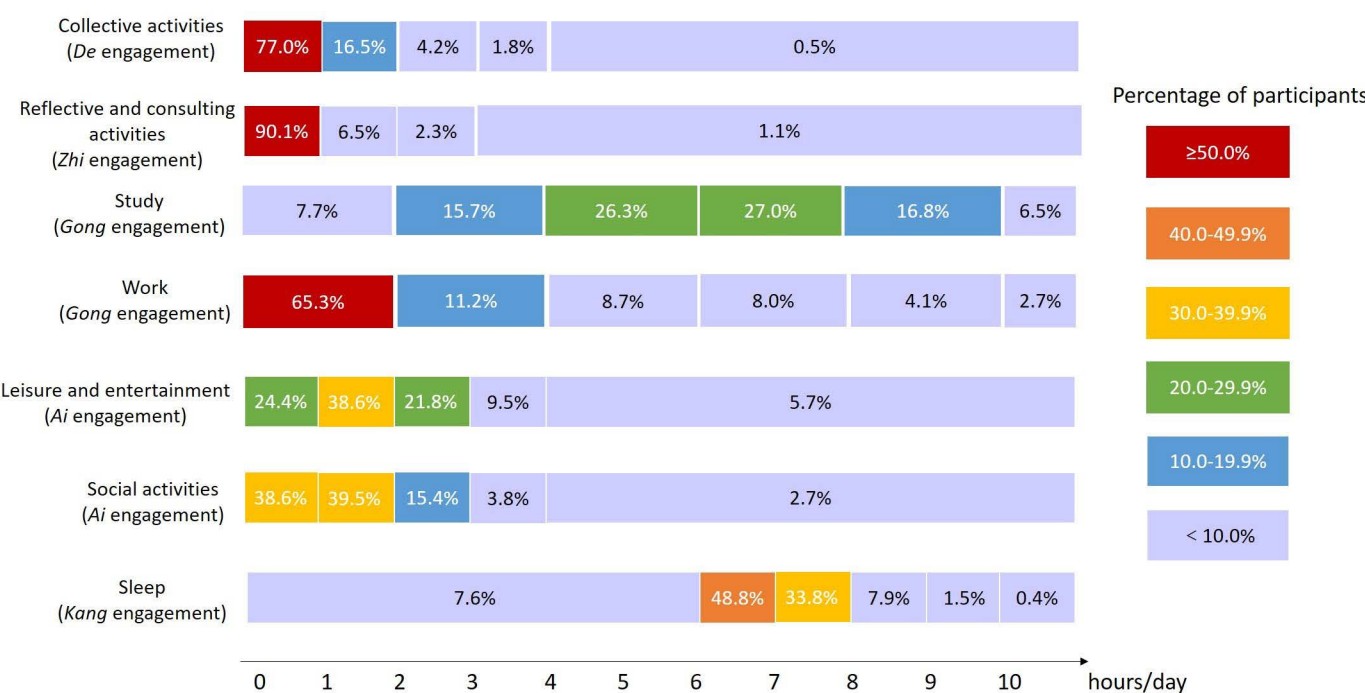

**Fig 2. Percentage of Participants' Average Time-Use in Five-Dimensional Occupational Engagement (N = 734).**

**Subtheme 1.1: Disequilibrium between mental and physical engagement.** Under the influence of COVID-19, excessive screen time and limited physical exercise caused an imbalance between mental and physical engagement. Three participants reported they were addicted to the Internet during the home isolation period, making them "feel guilty about wasting time".

**Table 3. Relationship between OHQ subjective subscale scores and DASS-21 scores (N = 734).**

| | Depression | | Anxiety | | Stress | |
|---|---|---|---|---|---|---|
| | r | *P value* | r | *P value* | r | *P value* |
| Level of occupational harmony | −.628 | <.001 | −.500 | <.001 | −.553 | <.001 |
| Equilibrium between two-sided occupational characteristics | −.459 | <.001 | −.395 | <.001 | −.465 | <.001 |
| Physical and mental engagement | −.398 | <.001 | −.331 | <.001 | −.397 | <.001 |
| Quiet and active engagement | −.404 | <.001 | −.364 | <.001 | −.425 | <.001 |
| Individual and social engagement | −.410 | <.001 | −.333 | <.001 | −.397 | <.001 |
| Five-dimensional occupational engagement | −.624 | <.001 | −.482 | <.001 | −.525 | <.001 |
| *De* engagement | −.453 | <.001 | −.351 | <.001 | −.377 | <.001 |
| *Zhi* engagement | −.435 | <.001 | −.342 | <.001 | −.333 | <.001 |
| *Gong* engagement | −.490 | <.001 | −.367 | <.001 | −.390 | <.001 |
| *Ai* engagement | −.439 | <.001 | −.338 | <.001 | −.359 | <.001 |
| *Kang* engagement | −.599 | <.001 | −.473 | <.001 | −.554 | <.001 |
| Multiple levels of human-environment transactions | −.516 | <.001 | −.398 | <.001 | −.449 | <.001 |
| Small-system | −.349 | <.001 | −.312 | <.001 | −.339 | <.001 |
| Medium-system | −.260 | <.001 | −.209 | <.001 | −.213 | <.001 |
| Large-system | −.341 | <.001 | −.239 | <.001 | −.301 | <.001 |
| Chronosystem | −.426 | <.001 | −.315 | <.001 | −.331 | <.001 |

OHQ: The Occupational Harmony Questionnaire, DASS-21: The Depression Anxiety Stress Scale.

Four participants spoke about "feeling a lack of exercise" due to excessive workload or the home quarantine policy. For example, participant F stated that "I was reading the literature every day, which made me become silly... I was afraid that my body could not endure this".

**Subtheme 1.2: Disequilibrium between active and quiet engagement.** Excessive engagement in the study or entertainment and lack of sleep were common problems identified by the participants. Five participants reported, "staying up late had become a habit" and "often went to bed at 3-4 AM". Some of them stayed up late to study, which made them feel "irritable", "anxious" and even "despairing". Others spent the night watching TV shows or reading novels, which made them lack the energy and motivation to study and work and caused depressive and anxiety symptoms.

**Subtheme 1.3: Disequilibrium between individual and social engagement.** Due to the COVID-19 protective measures, many participants reported a lack of social engagement, making them feel "bored", "lonely", and even "annoyed". Six participants complained about not being able to hang out or travel. For example, participant B said, "I was annoyed about being confined at home and unable to participate in other activities and meet other people". Five participants reported "feeling lonely" as they were very eager to communicate with their peers.

**Theme 2: Five-dimensional occupational engagement and mental health.** In the MOHar, it was asserted that occupational harmony, or a satisfactory life, was achieved through harmonious engagement in the five dimensions of occupations [15]. The five dimensions of occupational engagement have mutually enhancing and restricting relationships. The collective compatibility of the configuration of occupations is essential for mental health.

Among the 15 participants, 12 of them expressed dissatisfaction with their current life. Three types of common occupational disharmony patterns among participants were identified as shown in Fig 3.

**Subtheme 2.1: Excessive *Gong* engagement and insufficient *Kang* engagement.** Five participants demonstrated excessive *Gong* engagement, which directly restricted their *Kang*

**Table 4. Differences in DASS-21 Scores of university students with different time-use in five-dimensional occupational engagement (N = 734).**

| Occupational engagement | Depression subscale score | | Anxiety subscale score | | Stress subscale score | |
|---|---|---|---|---|---|---|
| | M (P25, P75) | P value | M (P25, P75) | P value | M (P25, P75) | P value |
| Collective activities (*De* engagement) | | .154 | | .749 | | .145 |
| 0 h/d | 3.0 (2.0, 7.0) | | 4.0 (2.0, 6.0) | | 6.0 (3.0, 9.0) | |
| <1 h/d | 3.0 (1.0, 6.0) | | 3.0 (2.0, 6.0) | | 6.0 (3.0, 7.0) | |
| 1-2 h/d | 3.0 (1.0, 6.0) | | 3.0 (2.0, 6.0) | | 6.0 (3.0, 7.5) | |
| 2-3 h/d | 4.0 (3.0, 7.0) | | 5.0 (2.0, 7.0) | | 7.0 (3.0, 8.0) | |
| ≥3 h/d | 2.0 (0.0, 7.0) | | 3.0 (1.0, 6.5) | | 4.0 (0.0, 7.0) | |
| Reflective and consulting activities (*Zhi* engagement) | | .002 | | .650 | | .260 |
| 0 h/d | 4.5 (2.0, 7.0) | | 4.0 (2.0, 6.0) | | 6.0 (4.0, 8.0) | |
| <0.5 h/d | 3.0 (1.0, 7.0) | | 3.0 (2.0, 6.0) | | 6.0 (3.0, 8.0) | |
| 0.5-1 h/d | 3.0 (1.0, 6.0) | | 4.0 (2.0, 6.0) | | 6.0 (3.0, 7.0) | |
| 1-2 h/d | 3.0 (0.0, 4.0) | | 3.0 (1.0, 5.8) | | 5.0 (2.0, 8.0) | |
| ≥2 h/d | 3.0 (1.0, 7.5) | | 3.0 (1.0, 7.0) | | 5.0 (3.0, 8.0) | |
| Study (*Gong* engagement) | | <.001 | | <.001 | | <.001 |
| <2 h/d | 7.0 (2.3, 11.0) | | 5.5 (3.0, 9.0) | | 8.0 (5.0, 11.0) | |
| 2-4 h/d | 5.0 (1.0, 7.0) | | 4.0 (2.0, 7.0) | | 6.0 (2.0, 8.0) | |
| 4-6 h/d | 3.0 (1.0, 6.0) | | 3.0 (2.0, 6.0) | | 5.0 (3.0, 7.0) | |
| 6-8 h/d | 3.0 (1.0, 6.0) | | 3.0 (2.0, 5.0) | | 5.0 (3.0, 7.0) | |
| 8-10 h/d | 3.0 (1.0, 5.0) | | 2.0 (1.0, 5.0) | | 5.0 (2.0, 7.0) | |
| ≥10 h/d | 4.0 (2.0, 7.0) | | 4.0 (3.0, 7.0) | | 7.0 (4.0, 9.8) | |
| Work (*Gong* engagement) | | .814 | | .049 | | .317 |
| <2 h/d | 3.0 (1.0, 7.0) | | 3.0 (2.0, 6.0) | | 6.0 (3.0, 8.0) | |
| 2-4 h/d | 3.0 (1.0, 7.0) | | 4.0 (1.8, 6.0) | | 6.0 (2.0, 7.0) | |
| 4-6 h/d | 4.0 (1.0, 6.0) | | 4.0 (2.0, 6.0) | | 5.0 (2.0, 7.8) | |
| 6-8 h/d | 3.0 (0.0, 7.0) | | 3.0 (1.0, 6.0) | | 6.0 (3.0, 9.0) | |
| 8-10 h/d | 3.0 (0.8, 6.0) | | 2.0 (1.0, 3.3) | | 4.5 (2.8, 6.3) | |
| ≥10 h/d | 3.5 (2.0, 7.0) | | 5.5 (3.0, 9.0) | | 7.5 (4.0, 9.8) | |
| Leisure and entertainment (*Ai* engagement) | | <.001 | | .003 | | .010 |
| 0 h/d | 5.0 (2.0, 8.0) | | 4.0 (2.0, 7.0) | | 7.0 (4.0, 10.0) | |
| <1 h/d | 3.0 (1.0, 5.0) | | 3.0 (1.0, 5.0) | | 5.0 (3.0, 7.0) | |
| 1-2 h/d | 3.0 (1.0, 6.0) | | 4.0 (2.0, 6.0) | | 6.0 (3.0, 7.0) | |
| 2-3 h/d | 4.0 (2.0, 7.0) | | 4.0 (2.0, 6.0) | | 6.0 (3.0, 8.0) | |
| 3-4 h/d | 4.5 (2.0, 8.0) | | 4.0 (2.0, 7.0) | | 7.0 (3.0, 8.3) | |
| ≥4 h/d | 5.5 (2.0, 9.0) | | 4.0 (1.0, 6.3) | | 7.0 (3.0, 10.3) | |
| Social activities (*Ai* engagement) | | .404 | | .126 | | .351 |
| 0 h/d | 5.0 (0.0, 7.0) | | 3.0 (2.0, 6.0) | | 5.0 (4.0, 8.0) | |
| <1 h/d | 3.0 (1.0, 7.0) | | 3.0 (1.0, 6.0) | | 6.0 (3.0, 7.0) | |
| 1-2 h/d | 3.0 (1.8, 6.0) | | 3.0 (2.0, 6.0) | | 6.0 (3.0, 8.0) | |
| 2-3 h/d | 4.0 (2.0, 7.0) | | 4.0 (2.0, 6.0) | | 7.0 (3.5, 8.0) | |
| 3-4 h/d | 3.0 (0.0, 7.8) | | 3.0 (1.0, 7.0) | | 5.5 (3.0, 8.8) | |
| ≥4 h/d | 4.0 (3.0, 6.0) | | 5.0 (3.0, 6.0) | | 6.0 (3.0, 7.0) | |
| Sleep (*Kang* engagement) | | .006 | | <.001 | | <.001 |
| <6 h/d | 6.0 (3.0, 8.0) | | 6.0 (3.0, 8.0) | | 7.0 (5.0, 9.0) | |
| 6-7 h/d | 3.0 (1.0, 6.0) | | 4.0 (2.0, 6.0) | | 6.0 (3.0, 8.0) | |
| 7-8 h/d | 3.0 (1.0, 6.0) | | 3.0 (1.0, 5.0) | | 5.0 (3.0, 7.0) | |
| 8-9 h/d | 3.0 (1.0, 6.0) | | 3.0 (1.0, 6.0) | | 5.0 (2.0, 8.0) | |
| ≥9 h/d | 4.0 (1.5, 9.3) | | 2.5 (1.0, 8.8) | | 4.5 (2.8, 12.3) | |

*(Continued)*

**Table 4.** (Continued)

| Occupational engagement | Depression subscale score | | Anxiety subscale score | | Stress subscale score | |
|---|---|---|---|---|---|---|
| | M (P25, P75) | *P* value | M (P25, P75) | *P* value | M (P25, P75) | *P* value |
| Rest (*Kang* engagement) | | <.001 | | .003 | | <.001 |
| 0-40 min/w | 4.0 (2.0, 8.0) | | 4.0 (2.0, 7.0) | | 7.0 (4.0, 9.0) | |
| 40-100 min/w | 3.0 (1.0, 6.0) | | 4.0 (2.0, 6.0) | | 6.0 (3.0, 8.0) | |
| 100-200 min/w | 3.0 (1.0, 6.0) | | 3.0 (2.0, 5.0) | | 5.0 (3.0, 7.0) | |
| ≥200 min/w | 2.0 (1.0, 6.0) | | 3.0 (1.0, 5.0) | | 5.0 (2.0, 7.0) | |
| Diet (*Kang* engagement) | | <.001 | | <.001 | | <.001 |
| Regular and healthy | 2.0 (0.0, 4.0) | | 2.0 (1.0, 4.0) | | 4.0 (2.0, 6.0) | |
| Healthy but irregular | 4.0 (2.0, 7.0) | | 4.0 (2.0, 7.0) | | 6.0 (3.0, 8.0) | |
| Regular but unhealthy | 3.0 (2.0, 7.0) | | 4.0 (2.0, 6.0) | | 6.0 (4.0, 8.0) | |
| Irregular and unhealthy | 6.0 (3.0, 9.0) | | 5.0 (3.0, 7.8) | | 7.0 (5.0, 10.0) | |
| Exercise (*Kang* engagement) | | .039 | | .011 | | .002 |
| 0 min/w | 4.0 (2.0, 7.0) | | 4.0 (2.0, 6.0) | | 6.0 (4.0, 8.0) | |
| 0-149 min/w | 3.0 (1.0, 6.5) | | 4.0 (2.0, 6.0) | | 6.0 (3.0, 8.0) | |
| ≥150 min/w | 3.0 (0.0, 6.0) | | 3.0 (1.0, 5.0) | | 4.0 (2.0, 7.0) | |

DASS-21: The Depression Anxiety Stress Scale

engagement (Pattern A in Fig 3). Specifically, these students were under great pressure to complete their coursework or research projects. Their daily life was very busy and they often "stayed up late to work on assignments". They lacked sleep and didn't have enough time for rest and exercise. Under excessive workload, these students were more likely to feel stressed, irritable, and even despairing.

**Subtheme 2.2: Insufficient *Zhi* and *Gong* engagement.** Five participants demonstrated insufficient *Zhi* engagement, which also negatively impacted their *Gong* engagement (Pattern B). The main problem with this group of students was that they had challenges in their studies that were difficult to solve by themselves. However, they seldom actively find solutions or consult with others, resulting in poor grades or slow research progress. The "self-isolated" status along with academic pressure caused depression, stress, and anxiety symptoms.

**Subtheme 2.3: Insufficient *De, Zhi,* and *Gong* engagement.** Two participants showed insufficient engagement in the *De* engagement, which negatively impacted their *Zhi* and *Gong* engagement (Pattern C). These students were not clear about their future goals and plans. They had difficulty with making choices and did not know where to put effort. They were not able to fully engage in their studies and work and thus lacked academic achievement, which made them feel confused and anxious about their future.

**Theme 3: Multiple levels of human-environment transactions and mental health.** The essence of occupational harmony was proposed to be harmonious human-environment transactions across multiple levels [15]. Four levels of systems were analyzed, which were small-system, medium-system, large-system, and chronosystem.

**Subtheme 3.1: Coherence with the small-system.** As most students stayed at home during the pandemic, the small system was usually composed of their home and family members. We found that participants' relationships with family members had a big impact on their moods. Five participants reported having arguments or conflicts with their parents, which made them feel displeased. In contrast, three participants had a more regular and enjoyable life at home than at school because of their good relationships with their families.

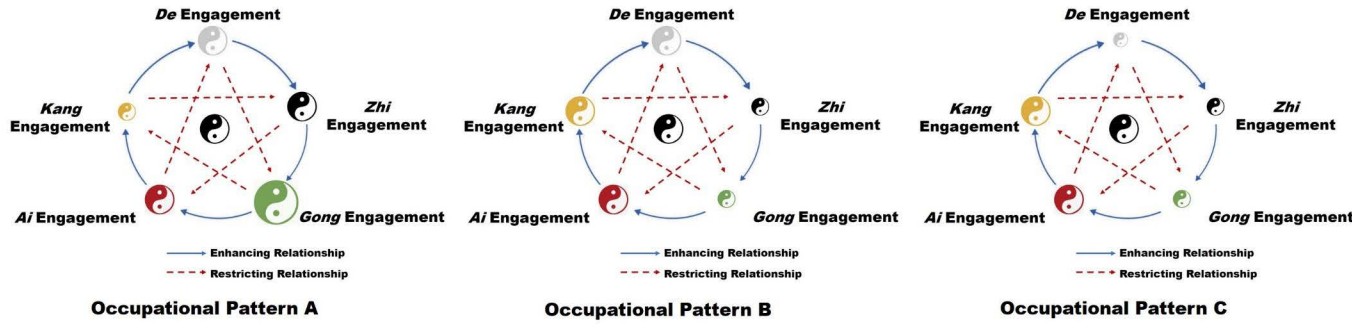

**Fig 3. Typical occupational disharmony patterns of participants (N = 15).** The size of Taiji diagrams indicates the level of occupational engagement. The Taiji diagram in the middle represents a moderate degree of occupational engagement. A smaller size indicates insufficient engagement, while a larger size indicates excessive engagement.

**Subtheme 3.2: Coherence with the medium-system.** For university students, the medium system mainly involves their schools. During the quarantine period, many students had to leave their schools and study on online learning platforms at home. Eight participants reported dissatisfaction with online learning because it was difficult to concentrate and communicate with teachers and classmates. The lack of a learning atmosphere made them "lack the motivation to learn and often slack off when taking online classes". Conversely, eight participants mentioned they felt stressed at school due to academic pressure.

**Subtheme 3.3: Coherence with the large-system.** Large systems include societies, countries, and the world. Due to the global pandemic, three participants' study abroad and exchange plans were disrupted. For example, student I had prepared for studying in Japan for a long time, but the outbreak of the pandemic "completely disrupted the original plan". When she recalled the experience, she said "I was so devastated...It was extremely hard for me to accept this fact...This was the biggest impact that the pandemic had on me."

**Subtheme 3.4: Coherence with the chronosystem.** Coherence with the chronosystem mainly refers to sufficient planning and management of time. Eight participants reported having no clear goals for life or lacking reasonable planning. One participant was "eager to graduate next year", but he did not have a plan for it and was also anxious and worried about meeting the graduation requirements. Three participants felt that life was "boring" and "meaningless" and did nothing every day.

## Discussion

This study investigated university students' mental health from the perspective of occupational harmony. The quantitative and qualitative results verified the primary hypotheses proposed by the MOHar. The results of the correlation analyses showed that the level of occupational harmony was positively correlated with QoL and negatively correlated with depression, anxiety, and stress, indicating that occupational harmony can lead to higher QoL and better mental health.

Although mental illness symptoms vary and the influencing factors are complex, the MOHar provides a theoretical framework to systematically analyze how occupational engagement impacts mental health. More than identifying the association relationships, we attempted to explain the causal relationships between occupational engagement and mental health based on the theoretical propositions of the MOHar.

According to the MOHar, it is believed that maintaining moderate engagement in occupations with different characteristics is essential for health and well-being. Disequilibrium

between the two-sided (*Yin* and *Yang*) characteristics of occupational engagement or disharmony among the five-dimensional occupational engagement will lead to mental illness [15]. The integration of quantitative and qualitative results indicates that students over-, under-, or even un-engaged in particular occupations are more likely to have mental illness symptoms, whereas moderate engagement is related to better mental health.

In terms of two-sided occupational characteristics, qualitative findings show that excessive mental engagement (e.g., excessive screen time) and insufficient physical engagement (e.g., lack of physical exercise), excessive active engagement (e.g., excessive study or entertainment) and insufficient quiet engagement (e.g., lack of sleep or rest), as well as insufficient social engagement (e.g., limited social activities), are related to mental health problems. There are studies that similarly found that the absence of physical exercise [23], heavy internet use (≥5 h/d) [24], poor sleep [25], and increased social isolation [26] were associated with stress, anxiety, and depression symptoms.

Regarding the five-dimensional occupational engagement, we found three common occupational disharmony patterns among university students, which were also supported by quantitative findings. The first pattern is characterized by excessive *Gong* engagement and insufficient *Kang* engagement, causing stress and anxiety symptoms. The Kruskal-Wallis *H* test shows that excessive *Gong* engagement (i.e., study more than 10h/d) is related to an increased level of anxiety, while insufficient *Kang* engagement (i.e., sleep less than 6h/d, rest less than 40min/w, unhealthy eating) is related to higher levels of depression, anxiety, and stress symptoms. The second pattern is characterized by insufficient *Zhi* and *Gong* engagement, causing depression, stress, and anxiety symptoms. Quantitative results indicate that insufficient *Zhi* engagement (i.e., reflective and consulting activities) contributed to a higher level of depression, and insufficient *Gong* engagement (i.e., studied less than 2h/d) is related to higher levels of depression, anxiety, and stress symptoms. The third pattern is characterized by insufficient *De*, *Zhi*, and *Gong* engagement, similar to the second pattern. Due to the representative activity for *De* engagement (i.e., collective activity) being different from the activities participants reported in the interview, we are not able to integrate the quantitative and qualitative findings for this pattern here. Future studies can further verify the relationship between life-planning activities and mental health.

Part of the above findings are consistent with previous literature. For instance, scholars found that difficulty with concentrating on academic work (*Gong* dimension) [26], disruptions to sleep and eating patterns (*Kang* dimension) [26] and lack of exercise (*Kang* dimension) [23], contributed to the increased levels of stress, anxiety, and depressive symptoms among students during the COVID-19 pandemic. To our knowledge, occupations in *De* and *Zhi* dimensions were seldom examined in previous studies, partially due to differences in cultural values and traditions. Additionally, the interrelationships among the five dimensions of occupation were revealed in the qualitative findings. For example, excessive *Gong* engagement can inhibit *Kang* engagement. Insufficient *De* engagement can lead to insufficient *Zhi* and *Gong* engagement. Analysis of these interrelationships is important for identifying key factors and thus informing intervention planning.

As Chinese culture is highly collectivist with a strong emphasis on group belonging and harmony [27], coherence across multiple levels of human-environment transactions is believed essential for mental well-being. Incoherence with the family (i.e., conflicts with family members), school (i.e., challenges of online learning, competitive pressure), social systems (i.e., maladaptation to COVID-19 measures), and the chronosystem (i.e., poor time management) were identified. Previous research also found that changes in the living environment [26] and poor family relationships [28] had an adverse effect on mental disorder symptoms. However, a paucity of studies explored environments from a multi-level lens and investigated

how social systems impact an individual's occupational engagement and mental health. Our findings reveal the mechanisms of how systems at different levels interact with each other and have a complex influence on an individual's mental health.

It is notable that according to the DASS-21 scores, 84.2% of the participants in this study were in normal mental health status. However, their OHQ results indicated that many university students with normal DASS-21 scores still had problems with occupational engagement. This is also supported by qualitative findings. Among the 15 participants, although the DASS-21 scores of 10 participants were within the normal range, they still had problems in different aspects of their life, such as study, career, family, and interpersonal relationships. These results suggest that the MOHar can provide a more holistic and functional perspective to studying university students' mental health.

Some limitations of this study should be acknowledged. Firstly, the study was conducted during the COVID-19 pandemic, which may limit the external validity of the results. Secondly, the survey was based on a convenience sample of university students with 70% female participants, which might result in selection bias and limited generalizability. Thirdly, to optimize the trade-off between a large number of everyday occupations and the limitation of questionnaire length, we selected some specific activities to represent five-dimensional occupational engagement. As these activities were proposed based on Chinese university students' life experiences and the study was conducted among Chinese students, the scope of transferability might be limited. Fourthly, the self-designed OHQ is not a standardized scale and its psychometric properties should be further tested.

Overall, the relationship between occupational harmony and mental health verified in this study provides a foundation for developing OT interventions. Future research can focus on developing interventions based on the MOHar to promote university students' mental health and test the efficacy and effectiveness of the intervention to bring about desired outcomes and explore the mechanisms that account for changes. This work, in turn, can lead to further theory development as it brings together research observations with interpretive reasoning proposed in the MOHar.

## Conclusion

This study examined university students' occupational engagement and mental health through the lens of the Model of Occupational Harmony. The findings indicated a positive relationship between that the level of occupational harmony and the mental well-being of university students. During the COVID-19 pandemic, key occupational factors affecting university students' mental health included excessive mental engagement and insufficient physical engagement, an imbalance between active engagement and quiet engagement, inadequate social engagement, and both excessive or insufficient *Gong* (productive) engagement, *De* (virtuous) engagement, *Zhi* (rational) engagement, and *Kang* (health maintenance) engagement. Additionally, incoherence with the family, school, social systems, and the chronosystem was observed. These insights are valuable for identifying the role that occupational therapy (OT) can play in promoting the mental health of university students during the pandemic and informing universities and relevant departments to implement more targeted mental health services.

The perspective of occupational harmony offers a functional and holist framework for examining mental health, which can identify individuals with mild mental health disorders but no clinical manifestations. This is helpful for screening university students with suboptimal mental health and conducting early interventions to prevent further development of more serious issues. An OT program aimed at enhancing occupational harmony may serve as a promising treatment option to promote mental health of university students.

The MOHar not only provides a tool for analyzing and addressing the mental health challenges faced by university students, but also has broader applications for individuals of different ages and with varying functional impairments. Currently, a research project is underway to evaluate the feasibility and effectiveness of applying the MOHar in patients with sleep disorders. Given that modern occupational therapy originated in Western societies and its development in China remains relatively preliminary, the MOHar offers a culturally-relevant theory to guide OT practice in China. Future studies should explore the efficacy and effectiveness of intervention programs based on the MOHar across different populations and contexts.

## Supporting information

**S1 Table. Themes and Subthemes Proposed by the Model of Occupational Harmony.** The two-sided, five-dimensional, and multi-level characteristics can be used to portray each occupational engagement and are not mutually exclusive. The characteristics are mainly determined by people's motivation of and subjective feelings about their engagement (i.e., characteristics of human consciousness that drive occupational engagement).
(DOCX)

**S2 Table. Profiles of 15 Participants in the Qualitative Study.** OHQ: The Occupational Harmony Questionnaire, DASS-21: The Depression Anxiety Stress Scale. Participant experience content areas. a: Characteristics of occupational engagement; b: Overall statement on mental health. Participants A to K participated in individual interviews. Participants L to O participated in a focus group discussion.
(DOCX)

## Author contributions

**Conceptualization:** Yijun Liu.

**Data curation:** Yijun Liu.

**Formal analysis:** Yijun Liu, Ruiqi She.

**Funding acquisition:** Yijun Liu.

**Investigation:** Yijun Liu, Ruiqi She.

**Methodology:** Yijun Liu.

**Project administration:** Yijun Liu, Ruiqi She.

**Resources:** Yijun Liu.

**Software:** Yijun Liu, Ruiqi She.

**Supervision:** Yijun Liu.

**Validation:** Yijun Liu.

**Visualization:** Yijun Liu.

**Writing – original draft:** Yijun Liu.

**Writing – review & editing:** Yijun Liu, Jin Xing.

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
