## [Decision Letter · Decision Letter 0]

6 Jan 2025

PONE-D-24-34209Analysis of university students' mental health from the perspective of occupational harmonyPLOS ONE

Dear Dr. Liu,

Thank you for submitting your manuscript to PLOS ONE. After careful consideration, we feel that it has merit but does not fully meet PLOS ONE’s publication criteria as it currently stands. Therefore, we invite you to submit a revised version of the manuscript that addresses the points raised during the review process.

We look forward to receiving your revised manuscript.

Kind regards,

Mohammad Sidiq, PhD Pain Sciences Physiotherapy

Academic Editor

PLOS ONE

Journal requirements: When submitting your revision, we need you to address these additional requirements. 1. Please ensure that your manuscript meets PLOS ONE's style requirements, including those for file naming. The PLOS ONE style templates can be found at https://journals.plos.org/plosone/s/file?id=wjVg/PLOSOne_formatting_sample_main_body.pdf and https://journals.plos.org/plosone/s/file?id=ba62/PLOSOne_formatting_sample_title_authors_affiliations.pdf. 2. Thank you for stating the following financial disclosure:  [Ethical approval was obtained from the ethics committee of Peking University First Hospital (Approval number: 2020Study104).].  Please state what role the funders took in the study.  If the funders had no role, please state: ""The funders had no role in study design, data collection and analysis, decision to publish, or preparation of the manuscript."" If this statement is not correct you must amend it as needed. Please include this amended Role of Funder statement in your cover letter; we will change the online submission form on your behalf.

Additional Editor Comments:

As per reviewers kindly do the revision and submit revised manuscript.

Reviewers' comments:

Reviewer's Responses to Questions

**Comments to the Author**

1. Is the manuscript technically sound, and do the data support the conclusions?

Reviewer #1: Yes

Reviewer #2: Yes

2. Has the statistical analysis been performed appropriately and rigorously? 

Reviewer #1: Yes

Reviewer #2: Yes

3. Have the authors made all data underlying the findings in their manuscript fully available?

Reviewer #1: Yes

Reviewer #2: Yes

4. Is the manuscript presented in an intelligible fashion and written in standard English?

Reviewer #1: Yes

Reviewer #2: Yes

5. Review Comments to the Author

Reviewer #1: The study is quiet good as the study clearly states that there was an influence of pandemic on the participants, then author have to highlight about the lacking in different section that occur due to the same.

Reviewer #2: The Model of Occupational Harmony (MOHar) provides a distinct and culturally relevant framework for examining mental health. The incorporation of traditional Chinese concepts into modern occupational therapy is admirable and innovative.

6. PLOS authors have the option to publish the peer review history of their article (what does this mean? ). If published, this will include your full peer review and any attached files.

**Do you want your identity to be public for this peer review?** For information about this choice, including consent withdrawal, please see our Privacy Policy .

Reviewer #1: No

Reviewer #2: No

---

## [Author Response · Author response to Decision Letter 1]

14 Jan 2025

We sincerely thank the academic editor and reviewers for their constructive feedback and valuable comments, which were of great help in revising the manuscript. Accordingly, the manuscript has been systematically revised and improved per their comments. Our responses to the reviewers’ comments are given below. All page numbers refer to the revised manuscript file with tracked changes.

Reviewers’ Comments to the Authors:

Reviewer 1

Comment 1: Add CONSORT flow chart for participants selection

Author Response 1: Thank you for this suggestion. We have added a flow chart of study population selection in the Participants section on page 7 lines 148-150.

Comment 2: Kruskal Wallis H test-Table for the same

Author Response 2: Table 4 on page 16 lines 291-292 and page 17-20 lines 293-294 demonstrates the results of Kruskal Wallis H test.

Comment 3: Meaningful excerpts- vocabulary correction

Author Response 3: As per the reviewer’s comments, we have changed the phrase “meaningful excerpts” to “meaningful quotes” on page 11 line 228.

Comment 4: Suggest the clinical significance of conducting this study and future scope for the same in the Conclusion section.

Author Response 4: As per the reviewer’s comments, we have added discussion of the clinical significance and future scope of conducting this study on page 29-30 lines 512-532.

Comment 5: Attach the file of Ethical Committee Certificate

Author Response 5: The file of Ethical Committee Certificate both in Chinese and English have been uploaded as attachments.

Comment 6: Add 1-2 group discussion video

Author Response 6: We have uploaded 2 audio recordings (1 individual interview and 1 focus group interview) and their transcripts both in Chinese and English.

Comment 7: Attach all measuring tools files entitled annexure section

Author Response 7: Thank you for this suggestion. All measuring tools both in Chinese and English were attached entitled annexure section.

Additional Clarifications

In addition to the above comments, we have attached the interview outline in the annexure section and refined expression in the Conclusion section on page 29 lines 501-512, page 30 lines 522-523 and lines 531-532.

---

## [Editor Report · Decision Letter 1]

22 Jan 2025

Analysis of university students' mental health from the perspective of occupational harmony

PONE-D-24-34209R1

Dear Dr. Liu

We’re pleased to inform you that your manuscript has been judged scientifically suitable for publication and will be formally accepted for publication once it meets all outstanding technical requirements.

Kind regards,

Mohammad Sidiq, PhD Pain Sciences Physiotherapy

Academic Editor

PLOS ONE

Additional Editor Comments (optional):

I would like to state that authors have addressed the comments raised by reviewers, and I feel we can proceed with the publication.
---

## [Editor Report · Acceptance letter]

PONE-D-24-34209R1

PLOS ONE

Dear Dr. Liu,

I'm pleased to inform you that your manuscript has been deemed suitable for publication in PLOS ONE. Congratulations! Your manuscript is now being handed over to our production team.

Kind regards,

on behalf of

Dr. Mohammad Sidiq

Academic Editor

PLOS ONE